

# Marker-assisted pyramiding of γ-tocopherol methyltransferase and glutamate formiminotransferase genes for development of biofortified sweet corn hybrids

Guihua Lv[1,*], Xiaolong Chen[1,*], Duo Ying[1], Jiansheng Li[2], Yinghu Fan[3], Bin Wang[4] and Ruiqiu Fang[1]

[1] Institute of Maize and Featured Upland Crops, Zhejiang Academy of Agricultural Sciences, Dongyang, Zhejiang, China
[2] National Maize Improvement Center of China, Beijing Key Laboratory of Crop Genetic Improvement, China Agricultural University, Beijing, China
[3] Chuxiong Academy of Agricultural Sciences, Chuxiong, China
[4] Institute of Vegetables, Zhejiang Academy of Agricultural Sciences, Hangzhou, Zhejiang, China
[*] These authors contributed equally to this work.

## ABSTRACT

Micronutrients, including vitamins, minerals, and other bioactive compounds, have tremendous impacts on human health. Much progress has been made in improving the micronutrient content of inbred lines in various crops through biofortified breeding. However, biofortified breeding still falls short for the rapid generation of high-yielding hybrids rich in multiple micronutrients. Here, we bred multi-biofortified sweet corn hybrids efficiently through marker-assisted selection. Screening by molecular markers for vitamin E and folic acid, we obtained 15 inbred lines carrying favorable alleles (six for vitamin E, nine for folic acid, and three for both). Multiple biofortified corn hybrids were developed through crossing and genetic diversity analysis.

## INTRODUCTION

Micronutrients (vitamins and minerals) are essential to people's health (*Farré et al., 2014*). At present, billions of people (mainly in developing countries) still suffer from "hidden hunger" due to insufficient intake of micronutrients (*Muthayya et al., 2013*). In the past 20 years, biofortification, enhancement of the levels of micronutrients in food crops through agricultural technologies, has been used as an important strategy to produce healthier food (*Garg et al., 2018*; *Saltzman et al., 2017*).

Sweet corn (*Zea mays* L. var. saccharata), a type of maize with high levels of sugar, is an invaluable source of protein, calories, essential fatty acids, vitamins, and minerals for human nutrition (*Wu et al., 2020*). However, the content of micronutrients present in the different sweet corn varieties varied significantly. The wide variability for micronutrient

Corresponding authors
Bin Wang, wb5682561@163.com
Ruiqiu Fang,
fangruiqiu2013@163.com,
fangrq@zaas.ac.cn

content in sweet corn unveils the great prospect of developing biofortified sweet corn varieties. Many quantitative trait loci (QTL) associated with micronutrients content have been identified (*Baseggio et al., 2019*; *Baseggio et al., 2020*; *Diepenbrock et al., 2017*; *Hershberger et al., 2021*; *Lone et al., 2021*; *Simic et al., 2012*; *Wu et al., 2022*). For example, *ZmVTE4*, encoding γ-tocopherol methyltransferase, is capable of catalyzing γ-tocopherol to α-tocopherol. α-tocopherol, the major constituent of vitamin E, shows the highest vitamin E activity (*Burton & Ingold, 1981*; *Kamal-Eldin & Appelqvist, 1996*). Two insertions in *ZmVTE4* promoter region and 5′ untranslated region (5′ UTR) affect the level of α-tocopherol through regulating gene expression (*Li et al., 2012*). Molecular markers (InDel7 and InDel118) corresponding to the two insertions were developed to screen for the favorable alleles. ZmCTM (catalysis of 5-M-THF to MeFox) functions as a key enzyme to convert 5-methyl-tetrahydrofolate (5-M-THF) to a pyrazino-s-triazine derivative of 4 α-hydroxy-5-methyl-tetrahydrofolate (MeFox) in folate metabolism. MeFox is the stable storage form of folic acid in seeds (*Goyer et al., 2005*). The natural asparagine-to-glycine substitution caused by an A-to-G single nucleotide variation in *ZmCTM* coding region enhances its enzymatic activity (*Zhang et al., 2016*). The G-allele can be identified by marker SNP682.

Commercial seeds of sweet corn are mostly $F_1$ hybrids, which are phenotypically superior and with significantly higher yield compared to their parents. Traditional corn breeding based on genetic crosses requires identifying the best parental combinations for creating elite hybrids. This process is very laborious, time-consuming, and cost ineffective. Moreover, the results are usually unpredictable and not always accurate. The level of genetic diversity between two parents has been proposed as a possible predictor of $F_1$ performance in crops (*Yousuf et al., 2021*). Accurate characterization of the genetic background of inbred lines can be very useful in selecting inbred lines for crossing (*Beckett et al., 2017*). The genetic variability can be assessed using agro-morphological traits, which may result in misleading estimates due to higher influence of environment on them. With the development of functional genomics and genome sequencing, marker-assisted selection has become an important approach for current crop improvement (*Nie et al., 2014*). Previous study established a core set of SSR molecular marker for characterizing genetic diversity of Chinese maize varieties and establishing the identity of new varieties (*Wang et al., 2011*).

Impressive progress has been made in biofortification of different elite crop inbred lines (*Prasanna et al., 2019*). There is an increasing demand for hybrid lines in practical production. Based on these requirements, we wondered whether genetic diversity together with favorable allele for vitamin E and/or folic acid could be analyzed to develop multi-biofortified sweet corns hybrids. Here, we obtained 15 inbred lines carrying favorable alleles through screening by molecular marker for vitamin E and folic acid (*Li et al., 2012*; *Zhang et al., 2016*). Together with the genetic diversity analysis (*Wang et al., 2011*), multiple biofortified corn hybrids were developed through crossing. This approach should greatly accelerate future biofortified breeding of sweet corn hybrids via effective selection of elite inbred lines with biofortification traits suit for optimal combination.

## MATERIALS AND METHODS

### Plant material

A set of 52 sweet corn inbred lines procured from different sources and maintained through selfing were taken for the study (Table S1). All these inbreds were planted in a randomized block design with two replications at the farmland of Zhejiang Academy of Agricultural Sciences (Dongyang, China) during 2020 and 2021.

### Genetic diversity analysis and allele screening

Genomic DNA was extracted using a modified CTAB extraction protocol (*Clarke, Moran & Appels, 1989*). The core 40 SSR primers were used for genetic diversity analysis (Table S2) (*Wang et al., 2011*). PCR amplifications were performed with a final reaction volume of 20 µL containing 30~40 ng genomic DNA. The PCR conditions were: 94 °C for 2 min, followed by 35 cycles of denaturation at 94 °C for 30 s, annealing at 55 °C for 30 s, extension at 72 °C for 30 s, and a last extension step at 72 °C for 10 minutes. The amplified products were resolved using 1.5% agarose gel or 12% PAGE (polyacrylamide gel electrophoresis) gel. Calculation of the PIC (polymorphism information content value) was based on the results obtained from SSR using the following formula: PIC $= 1 - \Sigma fi^2$, where $fi^2$ isthe frequency of the allele. A dendrogram were created using the unweighted pair group method using arithmetic averages (UPGMA) feature of NTSYS-pc software Version 2.2. InDel7 and InDel118 were used for *ZmVTE4* allele screening (*Li et al., 2012*), SNP682 was used for *ZmCTM* allele screening (Table 1) (*Zhang et al., 2016*). Primer sequences were obtained from the previously published paper by *Li et al. (2012)* and *Zhang et al. (2016)* with minor modifications.

### Quantification of free α-tocopherol

The endogenous free α-tocopherol contents were determined by Wuhan Greensword Creation Technology Co. Ltd. (Wuhan, China) based on UHPLC-MS/MS analysis. In brief, sample were frozen in liquid nitrogen, ground to fine powder, and extracted with 1.0 mL n-hexane at −20 °C for 12 h. After centrifugation (10,000 g, 4 °C, 20 min), the supernatants were collected and evaporated under mild nitrogen stream at 35 °C followed by re-dissolving in 100 µL ACN for UHPLC-MS/MS analysis (Thermo Scientific Ultimate 3000 UHPLC coupled with TSQ Quantiva; Thermo Fisher Scientific, Waltham, MA).

### Quantification of free folic acid

The endogenous free folic acid contents were determined by Wuhan Greensword Creation Technology Co. Ltd. (Wuhan, China) based on UHPLC-MS/MS analysis. In brief, sample were frozen in liquid nitrogen, ground to fine powder, and extracted with 1.0 mL 80% methanol aqueous solution at −20 °C for 12 h. After centrifugation (10,000 g, 4 °C, 20 min), the supernatants were collected and evaporated under mild nitrogen stream at 35 °C followed by re-dissolving in 100 µL 50% ACN for UHPLC-MS/MS analysis (Thermo Scientific Ultimate 3000 UHPLC coupled with TSQ Quantiva; Thermo Fisher Scientific, Waltham, MA).

**Table 1 Primers for InDel7, InDel118, and SNP682 used in this study.**

| Gene | Polymorphic site | Prime direction | Primer sequences (5′-3′) |
|---|---|---|---|
| *ZmCTM* | ZmCTM-CDS | Forward | TACGACGGTGGGTGTCAC |
| | | Reward | TGATAGGCGCTGGCATGATC |
| | ZmCTM-CDS2 | Forward | GTCATGCCTTGGATCGTGGG |
| | | Reward | ATGACGTCCTTACACAGCAC |
| *ZmVTE4* | ZmVTE4-InDel7 | Forward | TGCCGGCACCTCTACTTTAT |
| | | Reward | AGGACTGGGAGCAATGGAG |
| *ZmVTE4* | ZmVTE4-InDel118 | Forward | AAAGCACTTACATCATGGGAAAC |
| | | Reward | TTGGTGTAGCTCCGATTTGG |

## RESULTS AND DISCUSSION

To test the feasibility of the strategy, we analyzed the genetic diversity of 52 widely used sweet corn inbred lines using 40 pairs of SSR core markers (*Wang et al., 2011*). These markers produced 226 alleles, an average of 5.7 alleles per marker, suggesting a high frequency of allelic variation. The value of polymorphism information content (PIC) for each SSR locus varied between 0.27 and 0.87 with an average of 0.60. Based on the classification of PIC (PIC value < 0.25, low; 0.25 < PIC value < 0.5, intermediate; and PIC value > 0.5, high polymorphism) (*Botstein et al., 1980*), all the 40 SSR makers were found with moderate polymorphism and heterozygosity. The results suggested that these 40 SSR markers are suitable for assessing genetic diversity of sweet corn resources.

The dendrogram was obtained from the similarity coefficient and clustering was done by using the UPGMA algorithm with the NTSYS software program. The 52 inbred lines were divided into six distinct groups at the similarity coefficient level of 0.55 (Fig. 1). The first group accounted for 67.31% (35 inbred lines), the other groups were only for 32% (17 inbred lines). These results indicated that most of the inbred lines have similar genetic background.

We further analyzed the favorable alleles associated with vitamin E and folic acid content in 52 sweet corn inbred lines. ZmVTE4 and ZmCTM were identified to regulate biosynthesis of free α-tocopherol and folic acid content, respectively (Fig. 2A). Molecular markers (InDel7 and InDel118) corresponding to the two insertions in *ZmVTE4* promoter region and 5′ untranslated region (5′ UTR) were used to screen favorable allele for free α-tocopherol. Marker SNP682 in *ZmCTM* coding region was used to characterize alleles for free folic acid. Genotypic screening showed that there was 11.54% ($n = 6$) lines with deletion-allele in InDel7 and InDel118 loci, 17.31% ($n = 9$) lines with G-allele in SNP682 and 5.77% ($n = 3$) lines with both deletion-allele and G-allele (Fig. 2D). Our results revealed that most of the elite inbred lines used in breeding do not contain favorable alleles associated with vitamin E and folic acid content.

To develop hybrids with high level of vitamin E and folic acid, we chose inbred lines with micronutrients associated favorable alleles for crossing. Previous studies have suggested

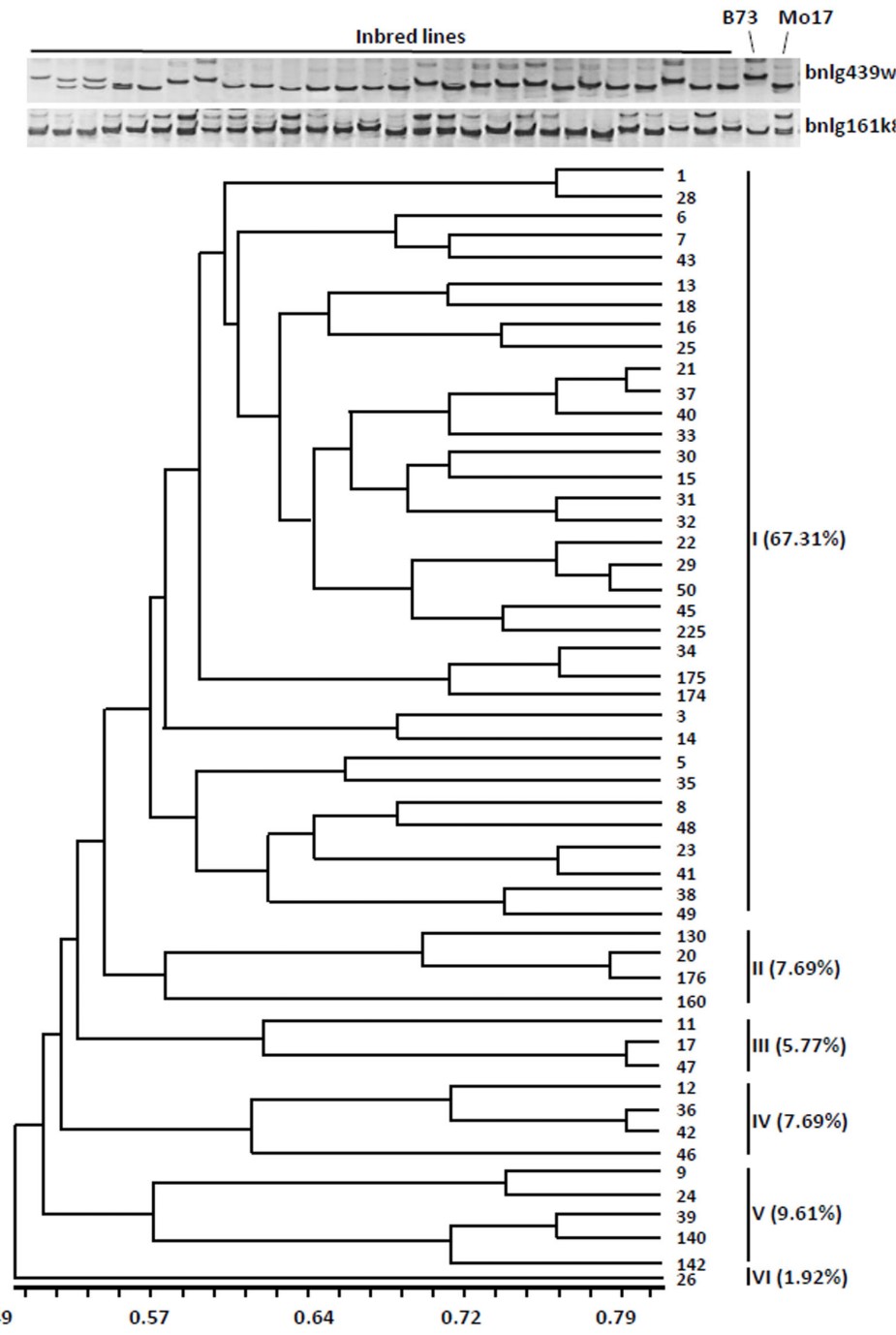

**Figure 1 Cluster dendrogram depicting genetic divergence among 52 inbreds based on 40 core molecular markers.** (A) Microsatellite polymorphism among sweet corn inbreds. (B) Cluster dendrogram depicting genetic divergence among 52 inbreds based on 40 core molecular markers.

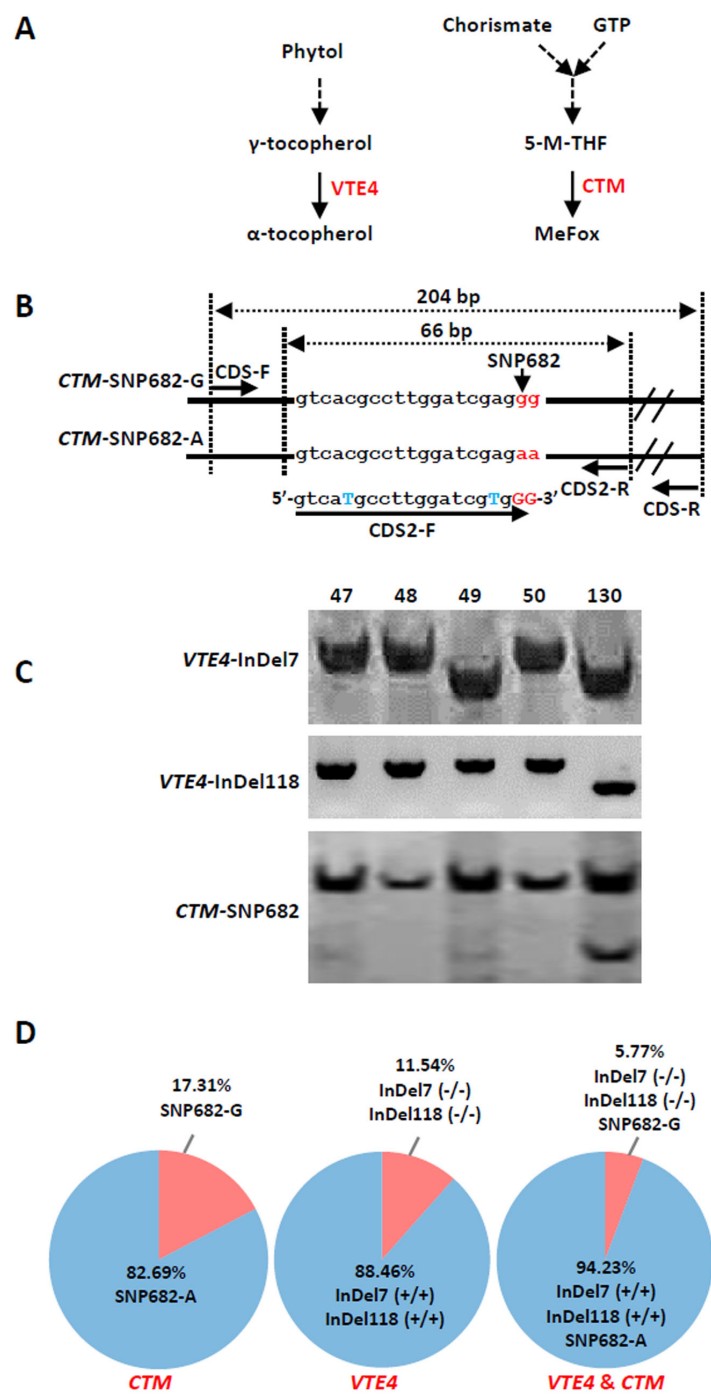

**Figure 2 Screening of favorable alleles for vitamin E and/or folic acid in the 52 inbred lines.** (A) Schematic of α-tocopherol and folate metabolism. VTE4, γ-tocopherol methyltransferase; 5-M-THF, 5-methyl-tetrahydrofolate; MeFox, a pyrazino-s-triazine derivative of 4 α-hydroxy-5-methyl-tetrahydrofolate; CTM, catalysis from 5-M-THF to MeFox. (B) Schematic illustration of SNP682 loci primer design, blue upper-case letters represent bases substituted to balance primer GC content of primer. (C) Representative pictures of allele assay at InDel7, InDel118, and SNP682 loci. 

**Figure 2 (…continued)**
(D) Analysis of allele at InDel7, InDel118, and SNP682 loci among 52 inbreds. InDel7 (+/ +), homozygous 7-bp insertion in the 5′ untranslated region (5′ UTR) of *ZmVTE4*; InDel7 (−/−), homozygous 0-bp insertion in the 5′ untranslated region (5′ UTR) of *ZmVTE4*; InDel118 (+/+), homozygous 118-bp insertion in the promoter region of *ZmVTE4*; InDel118 (−/−), homozygous 0-bp insertion in the promoter region of *ZmVTE4*; SNP682-G, homozygous G at position 682 in the coding sequence of *ZmCTM*; SNP682-A, homozygous A at position 682 in the coding sequence of *ZmCTM*.

that the level of genetic diversity between two parents could be used as a possible predictor of $F_1$ performance in crops (*Xiao et al., 1996*). Among the inbred lines carrying favorable alleles associated with vitamin E and folic acid content, lines with different genetic distance were selected to cross as parents. $F_1$ progenies from lines crosses with a large genetic distance (140 × 225, 142 × 225, 140 × 15, and 142 × 15) were observed with favorable agronomic traits (ear length, number of rows per ear, grain yield per main panicle, and 1,000-grain weight) (Figs. 3A–3C, Table 2). Notably, the highest yield per plant (272.58 g) was hybrid 140 × 225. In contrast, $F_1$ progenies of lines from same group had poor agronomic traits (Figs. 3A–3C, Table 2). The same trend can be found for other hybridization combination.

Meanwhile, we measured free α-tocopherol and folic acid in $F_1$ progenies. Based on the allele analysis, we found that hybrid 140 × 15 and hybrid 140 × 225 contains α-tocopherol favorable allele (InDel7$^{+/−}$InDel118$^{+/−}$ for hybrid 140 × 15 and InDel7$^{−/−}$InDel118$^{−/−}$for hybrid 140 × 225). Insertion in InDel7 and InDel118 loci affect the expression of *ZmVTE4* (*Li et al., 2012*). Quantification of free α-tocopherol (main component of vitamin E) revealed that the concentration in hybrid 140 × 15 and hybrid 140 × 225 was lower than that in hybrid 20 × 15 carrying no α-tocopherol favorable allele (InDel7$^{+/+}$InDel118$^{+/+}$) (Fig. 4). A similar variation pattern was observed for free folic acid in sweet corn kernel. The asparagine-to-glycine substitution caused by an A-to-G single nucleotide variation (SNP682) in maize *ZmCTM* enhances its enzymatic activity (*Zhang et al., 2016*). Homozygous G (SNP682$^{G/G}$) carrying hybrid 140 × 15 and 20 × 39 had significantly higher levels of free folic acid than heterozygous G/A (SNP682$^{G/A}$) carrying hybrid 140 × 15 in kernel (Fig. 4). In addition, there are differences between hybrid 140 × 15 and 20 × 39. Folates are unstable compounds, susceptible to oxidative and photo-oxidative catabolism (*Blancquaert et al., 2014*). Vitamin E is a potent antioxidant in plants, widely used to increase the shelf life of β-carotene in foods (*Choe & Min, 2009*). High level of α-tocopherol in Hybrid 140 × 15 may enhance folate stability. Our results demonstrated the validity of the strategy and provided supporting evidence for the notion that *ZmVTE4* (*Li et al., 2012*) and *ZmCTM* (*Zhang et al., 2016*) are key for the regulation of vitamin E and folic acid level in maize kernel.

## CONCLUSION

It is known that molecular marker-assisted selection is used in crop breeding (*Jena & Mackill, 2008*). Given that most commercial seeds are hybrids, we envisage that the

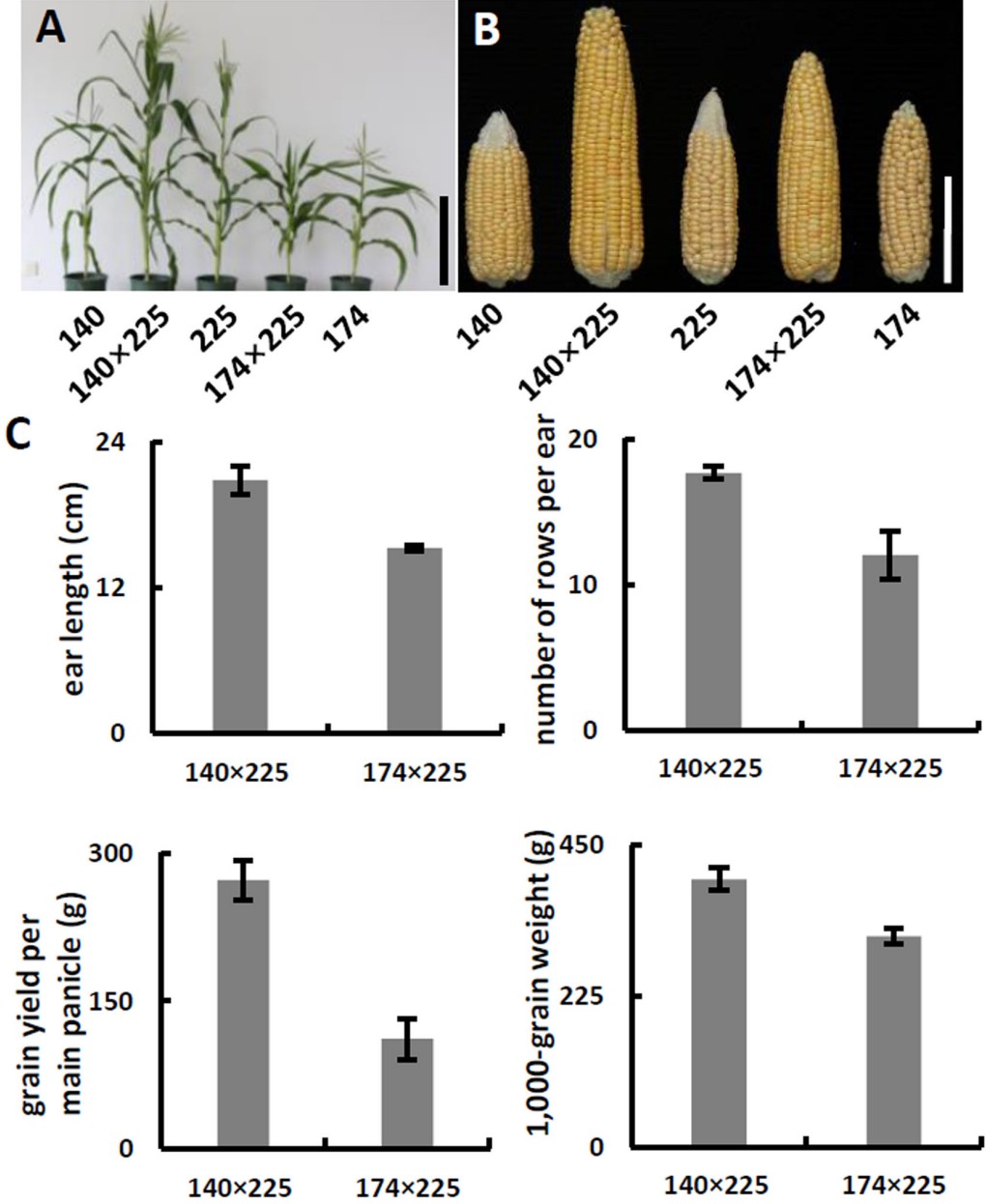

**Figure 3 Phenotypes and agronomic traits of parental inbreds and hybrids.** (A) Plant phenotype of parental inbreds and hybrids. Bar, 30 cm. (B) Phenotype of parental inbreds and hybrids on ears. Bars, 10 cm. (C) Analysis of agronomic traits hybrid 140 × 225 and hybrid 174 × 225. Error bars represent s.d.

**Table 2 Characterization of agronomic traits of hybrids.** Note, different letters show significant differences among treatment combinations at 5 probability level using Duncans multiple range test.

| Hybrid F1 | Growth phases (days) | Plant height (m) | Ear length (cm) | Number of rows per ear | 1,00-grain weight (g) | Grain yield per main panicle (g) | Sucrose content (mg/g) |
|---|---|---|---|---|---|---|---|
| 15 × 20 | 86 | 1.75 ± 4.24c | 16.17 ± 0.97gh | 15 ± 2.16abc | 47.27 ± 3.96a | 203.77 ± 25.1bcd | 172.39 ± 17.72b |
| 15 × 28 | 90 | 2.38 ± 0.02a | 19.77 ± 0.33bc | 15 ± 0.82abc | 34.31 ± 1.85de | 245.79 ± 7.49cd | 148.59 ± 13.72bcde |
| 20 × 39 | 86 | 1.65 ± 1.25cd | 18.23 ± 0.63def | 16.67 ± 2.49ab | 48.59 ± 2.24a | 231.24 ± 36.95abc | 118.53 ± 24.05 |
| 140 × 15 | 92 | 2.32 ± 0.02ab | 20.73 ± 0.45ab | 15.67 ± 0.47abc | 45.04 ± 2.85a | 263.29 ± 24.72a | 126.18 ± 0.97ef |
| 142 × 15 | 92 | 2.28 ± 0.09ab | 18.97 ± 0.92cde | 17.33 ± 1.25ab | 35.22 ± 0.3cde | 199.14 ± 9.95bcd | 140.8 ± 7.82def |
| 140 × 142 | 89 | 2.36 ± 0.06a | 17.5 ± 0.82efg | 14 ± 2.83abc | 36.11 ± 1.06bcd | 158.08 ± 13.49de | 144 ± 15.49cde |
| 140 × 174 | 88 | 2.28 ± 0.07ab | 17.93 ± 0.19def | 14.67 ± 0.47abc | 33.23 ± 0.74de | 159.11 ± 5.27de | 173.94 ± 4.33b |
| 174 × 175 | 87 | 1.69 ± 0.03cd | 17.4 ± 0.38fg | 15.33 ± 0.47abc | 34.31 ± 0.85de | 172 ± 10.71d | 172.55 ± 3.51b |
| 142 × 175 | 91 | 2.22 ± 0.04b | 19.37 ± 0.7bcd | 18.67 ± 0.47a | 33.36 ± 1.58de | 232.2 ± 11.52abc | 157.06 ± 4.85bcd |
| 39 × 225 | 89 | 2.34 ± 0.07a | 21.77 ± 0.39a | 17 ± 2.16ab | 38.92 ± 0.47bc | 261.75 ± 20.23a | 159.05 ± 7.84bcd |
| 140 × 225 | 89 | 2.31 ± 0.01ab | 20.83 ± 1.19ab | 17.67 ± 0.47ab | 39.92 ± 1.68b | 272.58 ± 20.23a | 167.1 ± 5.19bc |
| 142 × 225 | 90 | 2.28 ± 0.04ab | 21.4 ± 0.43a | 16.67 ± 1.7ab | 36.55 ± 1.3bcd | 237.5 ± 41.05ab | 163.12 ± 2.65bcd |
| 174 × 225 | 92 | 1.59 ± 0.03d | 15.23 ± 0.21h | 12 ± 1.63c | 31.41 ± 1.23e | 111.29 ± 20.79e | 205.52 ± 3.79a |

strategy used here will be widely adopted to accelerate biofortification breeding of various crops. The strategy allows for biofortification in elite F$_1$ hybrid with much higher efficiency and accuracy. A further improvement of this strategy could be achieved by integrating morphological traits assay to characterize genetic structure of parent lines comprehensively (*Mahato et al., 2018*). Further, the development of new polymorphic detection technologies such as KASP (*Semagn et al., 2014*), and whole-genome resequencing (*Jiao et al., 2012*; *Mace et al., 2013*) would also greatly expand the utility of this strategy. The strategy described here hold great promise to future biofortification breeding.

## ACKNOWLEDGEMENTS

We thank Baodong Cai from Wuhan Greensword Creation Technology Co. Ltd. for quantification of α-tocopherol and folic acid.

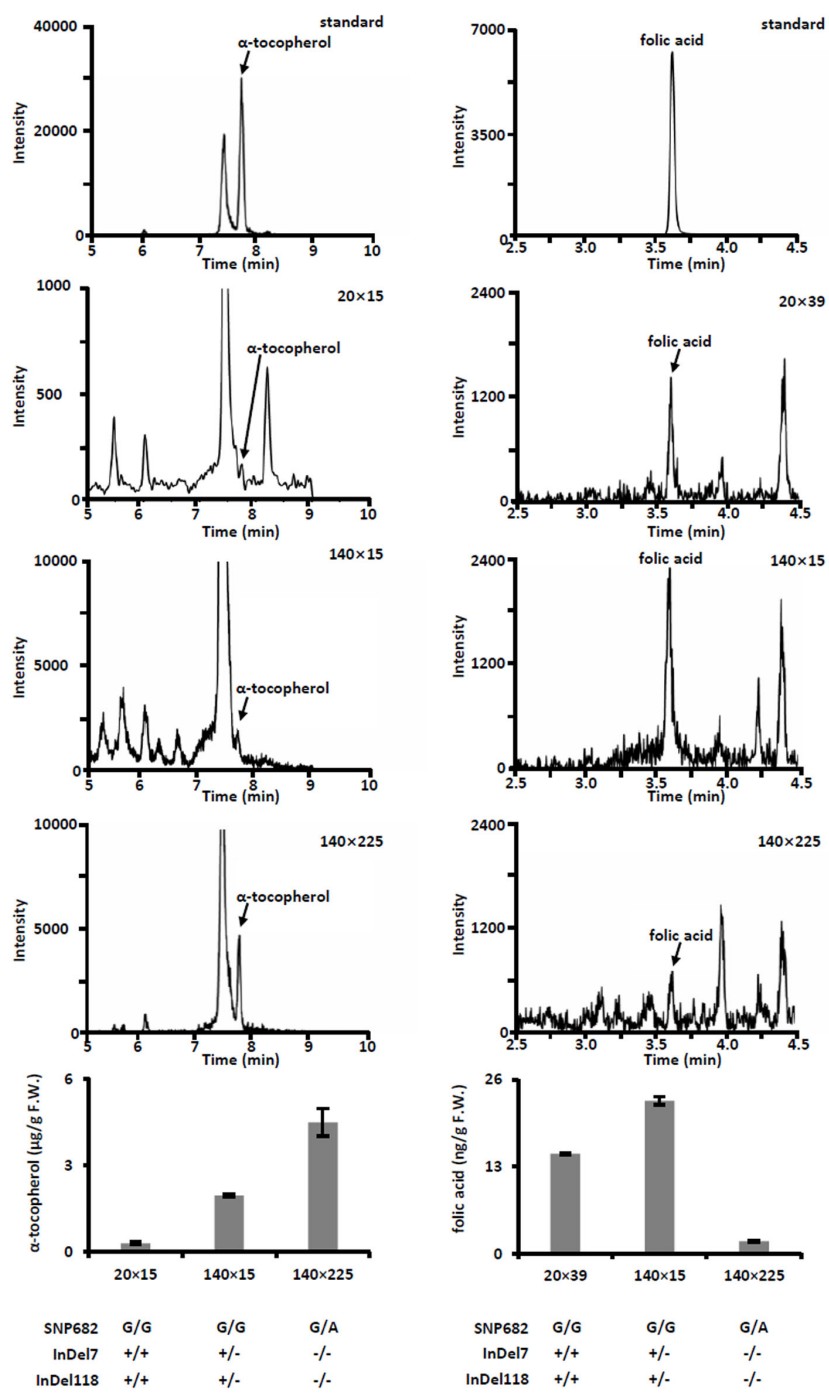

**Figure 4** **Quantification of free α-tocopherol and folic acid in kernel of hybrids.** Error bars represent s.d.

### Funding

This work was supported by the Zhejiang Provincial Natural Science Foundation of China under Grant No. LGN22C130014. The funders had no role in study design, data collection and analysis, decision to publish, or preparation of the manuscript.

### Grant Disclosures

The following grant information was disclosed by the authors:
Zhejiang Provincial Natural Science Foundation of China: LGN22C130014.

### Competing Interests

The authors declare there are no competing interests.

### Author Contributions

- Guihua Lv performed the experiments, prepared figures and/or tables, and approved the final draft.
- Xiaolong Chen performed the experiments, prepared figures and/or tables, and approved the final draft.
- Duo Ying analyzed the data, prepared figures and/or tables, and approved the final draft.
- Jiansheng Li conceived and designed the experiments, authored or reviewed drafts of the article, and approved the final draft.
- Yinghu Fan analyzed the data, prepared figures and/or tables, and approved the final draft.
- Bin Wang conceived and designed the experiments, authored or reviewed drafts of the article, and approved the final draft.
- Ruiqiu Fang conceived and designed the experiments, authored or reviewed drafts of the article, and approved the final draft.

### Data Availability

The raw measurements are available in the Supplementary Files.

### Supplemental Information

Supplemental information for this article can be found online at http://dx.doi.org/10.7717/peerj.13629#supplemental-information.

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
