# Peer review of "Marker-assisted pyramiding of γ-tocopherol methyltransferase and glutamate formiminotransferase genes for development of biofortified sweet corn hybrids"

_PeerJ, doi:10.7717/peerj.13629_

## Round 0.1 · original submission · Major Revisions

Authors should carefully revise the manuscript as indicated by the reviewers

·

Basic reporting

Abstract written is very poor. It should highlight the objectives and findings of your work, rather providing the background of your work.
Introduction part is well written and in line with your manuscript.
Materials and methods part is very poor. Provide the list of plant material used, parents used should be highlighted separately. Also give detailed information regarding PCR amplification like annealing temperature, duration of PCR cycles.

Experimental design

Results and discussion part is well elaborated, however, the references used are very old, provide some latest references. Lot of work has been done in the field of bio-fortification in recent years. It will be better to cite some latest references.
Provide the list of primers used in separate table and some Gel pictures to make your manuscript look more attractive.
Overall literature cited is very much old, add some recent references.

Validity of the findings

The language of the manuscript is good, however some grammatical mistakes have been highlighted. Little revision is needed to avoid the meaningless sentences.
The conclusion part is missing.
The manuscript written is very much informative and a good effort has been done by the authors to provide a way forward for enhancement of nutritional security through recent breeding procedures.

Additional comments

Line-28 Crops inbred lines (Correct the sentence like inbred lines in various crops)
Line-29 High-yield hybrid (Write high-yielding hybrids)
Line-30 sweet corn hybrid (Replace hybrid by hybrids)
Line-36 Keywords should be in alphabetical order
Line-44 a variety of maize (Write as a type of maize rather than a variety)
Line-74 Hybrids lines (Write as hybrids only or hybrid lines)
Line-77 Replace will by shall to a meaningful sentence
Use sweet corn hybrids everywhere instead of sweet corn varieties as your main focus is on hybrid development.
Scientific name of sweet corn is not correct, after L. write var.

·

Basic reporting

The research has technical significance starting from diversity screening, selection, crossing and biochemical and agronomic characterization.

The main paragraph from 159 to 173 requires a careful relook.
While it has been mentioned that 140/225 vs. 140/15 had single vs. Two alleles for two traits and accordingly the micronutrient.
1. What is the micronutrient concentration and allelic status of parents?
That should be mentioned and followed for each cross accomplished.

Several sentences need relook in terms of grammar and technicality

line-26, 46, 78-81, 135-136, 169-170.

Experimental design

Ok

Validity of the findings

Ok

Additional comments

NIl

Reviewer 3 ·

Basic reporting

The authors describe an experiment to introgress known genes to sweet corn via marker assisted selection. The title is misleading since this is not a new method. I would suggest reporting the title based on what the work consisted of. It looks to me like this was an assessment of genetic variability for the two traits, as well as an evaluation of the contribution of the two reported genes affecting vitamin E and folic acid.

The authors miss important references such as all the work previously done in sweet corn related to biofortification (vitamin A and E) (https://www.biorxiv.org/content/10.1101/2022.04.01.486706v1.abstract ; https://www.biorxiv.org/content/10.1101/2021.09.24.461734v1.abstract; https://acsess.onlinelibrary.wiley.com/doi/full/10.3835/plantgenome2018.06.0038 ; https://academic.oup.com/plcell/article/29/10/2374/6100443?login=true ; https://acsess.onlinelibrary.wiley.com/doi/full/10.1002/tpg2.20008).

Experimental design

The authors are not reporting a new method, but rather a single crossing cycle of lines selected based on marker-assisted selection. There is a brief discussion about heterosis, even though previous literature suggest the absence of heterotic groups in sweet corn. I would suggest the estimation of genetic parameters for the quantified traits, such as narrow sense and broad sense heritability.

Validity of the findings

The manuscript lacks quantitative genetics analysis to properly allow the discussion of heterosis. The work also lacks a proper description of the parental lines for other traits that are important for sweet corn. The two genes used for MAS had already been previously reported, so I am not sure what are the finding reported here.

---

## Round 0.2 · accepted · Accept

The revised version of the manuscript has been accepted for publication.

The SE added:

1. Please make minor grammatical corrections based on the attached PDF
2. Please add whether error bars represent s.e.m. or s.d. to the figure legend.